# Tumor-Targeting Peptides: The Functional Screen of Glioblastoma Homing Peptides to the Target Protein FABP3 (MDGI)

**DOI:** 10.3390/cancers12071836

**Published:** 2020-07-08

**Authors:** Abiodun Ayo, Eduard Figueras, Thomas Schachtsiek, Mazlum Budak, Norbert Sewald, Pirjo Laakkonen

**Affiliations:** 1Translational Cancer Medicine Research Program, Faculty of Medicine, University of Helsinki, 00014 Helsinki, Finland; abiodun.ayo@helsinki.fi; 2Organic and Bioorganic Chemistry OC III, Department of Chemistry, Bielefeld University, 33615 Bielefeld, Germany; figueraseduard@gmail.com (E.F.); T.Schachtsiek@uni-bielefeld.de (T.S.); mazlum.budak@uni-bielefeld.de (M.B.); norbert.sewald@uni-bielefeld.de (N.S.); 3Laboratory Animal Center, Helsinki Institute of Life Science (HiLIFE), University of Helsinki, 00014 Helsinki, Finland

**Keywords:** glioblastoma, FABP3, CooP, MST, alanine scan, phage display

## Abstract

We recently identified the glioblastoma homing peptide CooP (CGLSGLGVA) using in vivo phage display screen. The mammary-derived growth inhibitor (MDGI/FABP3) was identified as its interacting partner. Here, we present an alanine scan of A-CooP to investigate the contribution of each amino acid residue to the binding to FABP3 by microscale thermophoresis (MST) and surface plasmon resonance (SPR). We also tested the binding affinity of the A-CooP-K, KA-CooP, and retro-inverso A-CooP analogues to the recombinant FABP3. According to the MST analysis, A-CooP showed micromolar (K_D_ = 2.18 µM) affinity to FABP3. Alanine replacement of most of the amino acids did not affect peptide affinity to FABP3. The A-CooP-K variant showed superior binding affinity, while A-[Ala^5^]CooP and A-[Ala^7^]CooP, both replacing a glycine residue with alanine, showed negligible binding to FABP3. These results were corroborated in vitro and in vivo using glioblastoma models. Both A-CooP-K and A-CooP showed excellent binding in vitro and homing in vivo, while A-[Ala^5^]CooP and control peptides failed to bind the cells or home to the intracranial glioblastoma xenografts. These results provide insight into the FABP3–A-CooP interaction that may be important for future applications of drug conjugate design and development.

## 1. Introduction

One of the challenges in cancer therapy is to overcome the lack of specificity and selectivity of most antitumor agents. It is, therefore, necessary to modify anticancer agents to selectively target tumor cells without affecting the healthy tissues. Proteins such as p32 [1], FABP3 (also known as mammary-derived growth inhibitor (MDGI) [2]), and GRP78 [3] that are normally intracellular are often found on the cell surface in tumors. Thus, these proteins constitute functional molecular targets in cancer. As a treatment strategy, antibodies or peptides have become versatile agents to selectively deliver toxins to the tumor site [4,5]. Tumor-targeting peptides can be conjugated as delivery vehicles to active agents such as imaging probes, nanoparticles, or antitumor toxins to enhance tumor imaging and therapeutic targeting. These targeting peptides are either naturally occurring or ligand/ligand mimicking sequences that can be identified for instance by phage display screens [6,7,8].

We recently identified a linear glioblastoma-targeting nonapeptide (CooP; CGLSGLGVA) using an in vivo phage display screen [2]. CooP binds to the mammary-derived growth inhibitor/fatty acid binding protein 3 (FABP3) in the glioblastoma cells and its associated vasculature [2]. CooP peptide has been successfully used for the targeted delivery of chemotherapy [2] and different nanoparticles [9,10]. FABP3 is a small 15-kDa protein that has been reported to mediate fatty acid uptake [11] with the highest binding affinity to the polyunsaturated fatty acids (PUFAs) [12]. However, neither information about the CooP binding site in the FABP3 nor information about the essential amino acids for the binding exist.

Several sequence algorithm-based assays using the Basic Local Alignment Search Tool (BLAST, https://blast.ncbi.nlm.nih.gov/Blast.cgi), in silico digest or alanine scanning analysis have been previously used to identify active amino acid residues of peptides and proteins [13,14,15,16]. We envisioned in this study that a higher binding affinity may improve peptide’s cell penetrating capacity and efficient delivery of imaging agents to the tumor site.

Alanine scanning is a conventional approach exploited to understand the effect of the systemic replacement of each amino acid on the peptide sequence [17,18,19]. These provide information on the peptide’s activity after interaction studies with the target protein using binding techniques such as Surface Plasmon Resonance (SPR [20]) and Microscale Thermophoresis (MST [21]). MST is a biophysical technique used for the analysis of biomolecular interactions based on fluorescence of one of the interacting partners (autofluorescence or covalently attached fluorophores). The binding events occur when there is a change in thermophoresis, a phenomenon known to occur when the fluorescent molecules in solution move along an infrared (IR)-laser-induced temperature gradient irrespective of the molecular size, size changes, hydration shell, and other physical properties upon binding [22,23,24].

In this study, we performed alanine scan and microscale thermophoresis (MST) studies to reveal the amino acids necessary for binding of the CooP derivatives to its target protein FABP3. Furthermore, we used both in vitro and in vivo glioblastoma models to validate the FABP3-dependent binding of fluorescently labeled CooP and its analogues.

## 2. Results

### 2.1. Cloning, Expression, and Purification of Recombinant Human FABP3 Protein

The gene encoding human *FABP3* was cloned into a bacterial expression vector to produce a recombinant *N*-terminal His-tagged fusion protein for peptide binding assays. The expression level of the His_6_-FABP3 protein was verified using an anti-His-tag antibody (Figure 1A). The recombinant His-FABP3 was subjected to two consecutive purification steps: (i) immobilized metal affinity chromatography (IMAC) (Figure 1B) and (ii) size exclusion chromatography (SEC) (Figure 1C). IMAC-purified protein was analyzed by using Bradford solution, and protein concentration was determined by nanodrop (BioSpec nano spectrophotometer). A total yield of 54 mg/mL IMAC-purified His_6_-FABP3 was obtained (Figure 1B: Lanes 4–8). A fraction of the IMAC-purified protein (21 mg/mL) corresponding to lane 4 (Figure 1B) was subjected to a further purification step by SEC for improved purity of the recombinant protein. SEC showed two peaks, one of which corresponds to the purified FABP3 (Figure 1D, blue peak) and the other one to impurities including the buffer exchange, imidazole, salts, etc. (Figure 1D, yellow peak).

### 2.2. Alanine Scanning Analyses and Synthesis of TAMRA-Labeled Selected CooP Variants

An alanine scan was performed to assess the role of the different amino acids in the CooP peptide on binding to recombinant FABP3. An alanine residue was added to the *N*-terminus of the CooP peptide (A-CooP) as a spacer between the homing peptide and a cargo. Our group has previously reported that introduction of an alanine at the *N*-terminus does not negatively affect its binding [2]. Then, each amino acid residue was replaced by alanine to generate eight different A-CooP-Ala variants: A-[Ala^1^]CooP-A-[Ala^8^]CooP (Table 1). Also, the native carboxylic acid at the *C*-terminus was replaced by an amide group. We also investigated the influence of further modifications at the *N*- or *C*-termini of A-CooP by introducing a lysine residue (KA-CooP and A-CooP-K). Finally, we synthesized a retro-inverso version of the peptide (RI-A-CooP), which could have similar binding properties to the native CooP with improved protease stability (Table 1). For the synthesis of fluorescently labeled 5(6)-carboxytetramethylrhodamine (TAMRA)-conjugated selected A-CooP variants, *N*-terminus was modified by the addition of carboxyl-TAMRA dye and the *C*-terminus was modified by replacing carboxylic acid with an amide group (Figure 2).

### 2.3. Surface Plasmon Resonance Measurement of the FABP3-A-CooP Binding Affinity

For the surface plasmon resonance (SPR) measurements, A-CooP (ACGLSGLGVA) and the scrambled control (CVAALNADG) peptides were immobilized into the sample and reference flow cells, respectively, on a sensor CM5 chip surface. Then, binding of His_6_-FABP3 to the immobilized peptides was analyzed. After generation of the reference-corrected sensorgrams and determination of the binding parameters, the equilibrium dissociation constants (K_D_) and maximum response amplitude (R_max_) values were calculated from the steady state affinity measurements (Figure 3A and Appendix A). The mean calculated K_D_ and R_max_ values from three independent experiments for the FABP3-A-CooP were 16.7 µM and 25.36 RU, respectively (Figure 3B and Appendix A). This analysis verifies the binding of A-CooP to FABP3.

### 2.4. Microscale Thermophoresis Analyses

For the binding affinity measurements between the recombinant FABP3 and the A-CooP, its variants, and the control peptide, we used the microscale thermophoresis. Figure 4A shows the scheme for the Alanine scan of A-CooP variants. For these analyses, FABP3 was labeled with an NT-650-NHS fluorescence. Two-fold serially diluted peptides ranging from 500 µM to 0.0153 µM were titrated against a constant FABP3 concentration (20 nM). The binding events were generated following changes in fluorescence (ΔF_norm_) against peptide concentrations. Subsequently, the equilibrium dissociation constant (K_D_) was calculated using a 1:1 fit model from the law of mass action. The MST binding affinities (K_D_ values) are represented in a bar graph (Figure 4B). The smallest K_D_ value equals the best affinity, and the highest value is the lowest affinity. K_D_ values for A-[Ala^1^]CooP, A-[Ala^5^]CooP, and scrambled control peptides are not represented in the bar graph since they showed no binding to the FABP3. The changes in fluorescence (ΔF_norm_) were normalized to the bound fraction (Figure 4C). The data points towards 1 were considered bound fractions, while data points less than 1 represented the unbound fractions.

The MST affinity data showed a K_D_ value of 2.18 µM for the A-CooP peptide (Figure 5A), which was similar to that obtained with SPR (Figure 3 and Appendix A). According to the MST data, replacement of the cysteine by alanine in A-[Ala^1^]CooP totally abolished binding to FABP3 (Figure 5B) as did the replacement of glycine in position 5 (A-[Ala^5^]CooP) (Figure 5C). In addition, a single glycine replacement with alanine in position 7 (A-[Ala^7^]CooP) reduced the peptide binding to FABP3 by two orders of magnitude (K_D_ = 133 µM) (Figure 5D). On the other hand, the A-CooP-K peptide showed the highest binding affinity to the recombinant FABP3 with a K_D_ value of 0.07 µM (Figure 5E), which was about 30-fold higher than the binding affinity of the A-CooP (Figure 5A). The binding affinities of the A-[Ala^2^]CooP, A-[Ala^3^]CooP, A-[Ala^4^]CooP, A-[Ala^6^]CooP, A-[Ala^8^]CooP, and KA-CooP were similar, ranging between 0.11–0.63 µM (Figure 5F-K) while RI-A-CooP showed 10-fold lower binding affinity compared to A-CooP (Figure 5L). As anticipated, the control peptide did not bind to FABP3 (Figure 5M). Thus, in addition to the cysteine residue, the two glycines are required for the CooP binding to the recombinant FABP3.

### 2.5. Validation of the MST Data In Vitro

To validate the MST data by further cellular binding studies in vitro, the *N*-terminus of A-CooP-K (highest affinity), A-CooP, A-[Ala^5^]CooP (lowest affinity), and scrambled control peptide were fluorescently labeled with TAMRA. For this purpose, FABP3 was overexpressed as a green fluorescent protein (GFP)-fusion (FABP3-GFP) in the U87MG glioblastoma cells that show low endogenous FABP3 expression. GFP-expressing U87MG cells were used as a control [2]. FABP3 protein expression was verified by Western Blot analysis (Figure 6A). The cells were incubated with fresh peptide-containing growth medium for 60 min followed by fixation and preparation for microscopy. The A-CooP-K peptide showed high uptake by the FABP3-expressing cells, while the A-CooP peptide showed moderate uptake and A-[Ala^5^]CooP and control peptides did not show any binding or internalization (Figure 6B,C). Thus, in line with the MST data, A-CooP-K showed better binding to the FABP3-overexpressing U87MG cells than A-CooP (Figure 6B,C). Neither A-CooP-K nor A-CooP bound to or was internalized by the GFP-expressing U87MG cells that express low amounts of FABP3 (Appendix A), suggesting binding of A-CooP-K and A-CooP to be dependent on the expression of their target protein FABP3. No binding or internalization of A-[Ala^5^]CooP or the control peptide was detected in either cell line (Figure 6B,C and Appendix A).

### 2.6. A-CooP and A-CooP-K Peptides Home to Intracranial Glioblastoma Xenografts

To test the homing of the peptides to the intracranial glioblastoma xenografts, we grafted the GFP-expressing patient-derived BT12 glioblastoma cells intracranially into immunocompromised mice. After 25 days, fluorescently labeled peptides were infused intravenously into tumor-bearing animals and were allowed to circulate for 60 min. The brain tissue was collected and processed for immunofluorescence and imaging. Similar to the in vitro results, no homing of the control peptide (Figure 7A,B) or A-[Ala^5^]CooP (Figure 7C,D) was detected while both A-CooP (Figure 7E,F) and A-CooP-K (Figure 7G,H) homed to the intracranial tumors. The mean fluorescence intensity in the tumor sections was quantified using the cell profiler image analysis tool (https://cellprofiler.org). The white (4′,6-diamidino-2-phenylindole, DAPI) and green (tumor) images were combined to form the background output using the ImageMath (Add) module. The background was subtracted from the red (TAMRA) channel using the ImageMath (Subtract), module and the TAMRA peptide intensity was measured accordingly. Analyses were carried out on at least 20 histological sections from at least three animals per peptide group. Both the A-CooP-K and A-CooP peptides but not the A-[Ala^5^]CooP peptide showed significantly better homing compared to the control peptide (Figure 7I). Thus, both the in vitro binding and in vivo homing experiments validated the MST results.

## 3. Discussion

The glioblastoma homing peptide, CooP, was identified by using the in vivo phage display, and FABP3/MDGI was identified as its binding partner [2]. Here, we describe measurements of binding affinity between the A-CooP peptide and recombinant FABP3, reveal essential amino acids required for binding, and show improved affinity of the A-CooP-K variant towards FABP3, leading to improved glioblastoma targeting.

In this study, we analyzed binding of A-CooP and its 12 variants to recombinant FABP3. We aimed to understand the role of each amino acid in the A-CooP sequence on the binding to FABP3 in the context of glioblastoma targeting. By using alanine scanning, we generated A-CooP variants with each amino acid residue substituted by alanine residue. A-CooP-K, KA-CooP, and a retro-inverso A-CooP were also generated. Consequently, binding affinity measurements performed by microscale thermophoresis identified seven A-CooP variants (A-[Ala^2^]CooP, A-[Ala^3^]CooP, A-[Ala^4^]CooP, A-[Ala^6^]CooP, A-[Ala^8^CooP, KA-CooP, and A-CooP-K) that showed similar or better binding (K_D_ value ranging between 0.07–2.18 µM) to the target protein than the original A-CooP. Among those, the A-CooP-K peptide showed the highest binding affinity to FABP3. In addition, two variants (retro-inverso A-CooP and A-[Ala^7^]CooP) showed lower binding affinity than the original A-CooP. Two variants (A-[Ala^1^]CooP and A-[Ala^5^]CooP) did not bind to FABP3. Alanine variants are referred to briefly from now on as Ala (Ala^1^–Ala^8^) for further discussion purposes.

Alanine replacement of most of the amino acids in the A-CooP sequence except for Ala^1^, Ala^5^, and Ala^7^ did not affect peptide affinity to FABP3. The direction of binding curves and amplitude of MST signals could provide information on the binding orientation [25] and the interaction site [26,27] accordingly. Adjudging via the MST data, the binding curves of most of the variants follow the same direction and the MST signals showed similar response amplitudes ranging between 15–24RU. Similarly, with SPR, the maximum response amplitude (Rmax) measurement of FABP3 to A-CooP was 25.36RU. These suggest that the binding of FABP3 to A-CooP and the other variants alike follow the same binding orientation and site-specific binding. On the other hand, our analyses with Ala^1^, Ala^5^, and Ala^7^ revealed different binding curves and signal responses. Ala^1^ showed no binding and no MST signal response, as did the control scrambled peptide. Ala^5^ showed an opposite MST binding curve (with an ambiguous K_D_) and no signal response, indicating no binding to FABP3. Ala^7^ also showed an opposite MST binding curve (with low binding affinity, K_D_ = 133 ± 75 µM) and a high signal response (138RU), indicating a different binding orientation and off-site binding to FABP3.

A single replacement of cysteine with alanine at position 1 (Ala^1^) totally abolished the binding to FABP3. Cysteine is known to play a role in structural stability of cyclic peptides through the formation of disulfide bridge [28,29,30,31]. The original CX_7_C phage-displayed peptide library from which the CooP peptide was identified contained two cysteine residues which are important for the stabilization of the peptide structure [2,32]. A previous study showed that adding an extra cysteine residue to a cyclic nonapeptide, iRGD (CRGDK/RGPD/EC), improved the tumor penetrating function [33]. In addition, one cysteine residue may interact with cysteine(s) in other peptides to form peptide complexes [34]. It is also possible that the single cysteine residue in the original CooP sequence interacts with the cysteine residue (Cys^125^) on the FABP3 sequence to stabilize the complex.

In addition to cysteine, the two glycine residues substituted with alanine in positions 5 and 7 in the A-CooP sequence greatly impacted their binding affinities. In reference to the MST data, alanine in position 7 reduced binding to FABP3 by about 100-fold (K_D_ between 2.18 µM and 133 µM) while replacement of glycine with alanine in position 5 resulted in total loss of binding. It was interesting to note that the two glycine replacements could bring about a major change in the binding affinity of A-CooP to FABP3. In addition, Ala^5^ showed no detectable in vitro uptake or in vivo homing. Thus, the MST results predicted very well the peptide behavior in other assays. Based on the MST data, the in vitro cellular uptake, and in vivo glioblastoma homing studies, we conclude that the glycine residues in positions 5 and 7 are critical for the FABP3 binding.

As a result of the complex conjugation often associated with most peptide-based delivery systems, in addition to the alanine scan, we aimed to modify the peptide to facilitate the delivery of a variety of active agents such as imaging probes or anticancer toxins to tumor site without interfering with the peptide’s activity. We therefore envisioned an inclusion of a conjugation site that would not affect the *N*-terminus. For this reason, a lysine residue was included either at the *C*-terminus or the *N*-terminus. The 11-mer A-CooP variant, A-CooP-K, showed substantially improved FABP3 binding based on the MST and in vitro analyses. Addition of a lysine (K) residue to the *C*-terminus of A-CooP (A-CooP-K) increased the binding affinity of A-CooP from 2.18 µM K_D_ to 0.07 µM K_D_. Thus, modification at the *C*-terminus of A-CooP with lysine did not negatively affect peptide’s activity but rather showed 30-fold higher affinity towards FABP3. On the other hand, *N*-terminal modification of A-CooP with lysine did not affect the peptide’s binding (KA-CooP; 0.63 ± 0.13 µM).

Higher binding affinity of tumor-targeting peptides to their receptors could improve the delivery efficiency of active cargos to tumor sites [35,36]. Indeed, fluorescently labeled A-CooP-K also showed significant uptake in the U87MG glioblastoma cells overexpressing the FABP3-GFP fusion protein compared to U87MG expressing GFP, suggesting that the binding was FABP3 dependent. In addition, both A-CooP-K and A-CooP showed excellent homing to the intracranial patient-derived glioblastoma xenografts. We have previously reported peptide-mediated delivery of an anticancer drug chlorambucil conjugated to CooP and a cell penetrating peptide (ARF(1-22) peptide [37]) to glioblastoma xenografts [2]. As A-CooP-K shows even better homing ability to glioblastoma xenografts in vivo and appears to be internalized into the cells, the A-CooP-K peptide may not require an additional cell penetrating peptide for targeted delivery of therapies. A functional tumor-targeting peptide would allow efficient delivery and accumulation of various cargos in the tumor site thereby, reducing off-target effects [38]. Our approach suggested A-CooP-K as a novel theranostic tool for glioblastoma detection and treatment due to the 30-fold enhanced binding affinity towards target (FABP3) compared to the original A-CooP peptide.

## 4. Materials and Methods

### 4.1. FABP3 Cloning and Expression

The *N*-terminal His_6_-tagged human FABP3 construct was generated using the Genome Biology Unit cloning service, University of Helsinki (https://www.helsinki.fi/en/researchgroups/genome-biology-unit/clones-and-cloning). Briefly, FABP3 entry clone from the human ORFeome library was transferred into the pDEST17 destination vector (Invitrogen Life technologies, Helsinki, Finland) using the standard left right “LR” reaction protocol. The destination plasmid containing FABP3 was transformed into the BL21 (DE3) *Escherichia coli* bacterial strain, and expression was induced with 0.1 mM isopropylthiogalactoside (IPTG, Bioline Bio-37036, Memphis, TN, USA) after an OD_600_ of 0.6 was reached. The incubation temperature was adjusted to 25 °C, and cells were cultured overnight (with shaking at 150 rpm). Bacterial cells were centrifuged at 4200× *g* for 1 h, and the supernatants were discarded. The cell pellets were resuspended in phosphate buffer (50 mM Na_2_HPO_4_, 300 mM NaCl, pH 7.4) supplemented with 1 mM phenylmethanesulfonyl fluoride (PMSF) protease inhibitor (100 mM stock solution prepared in isopropanol, Sigma, Schnelldorf, Germany) and 20 µg/mL DNase l solution (Thermo Scientific, Helsinki, Finland) followed by cell lysis using French pressure cell press at 1000 psi or sonicator (Soniprep 150; MSE, London, UK) at 14 amplitude microns speed for 1 min run and 1 min pulse cycle for a 6-min total session. The generation of U87MG human glioblastoma cells stably expressing the FABP3/MDGI-GFP fusion protein has been described previously [2].

### 4.2. FABP3 Purification by Immobilized Metal Affinity Chromatography (IMAC)

Cell extracts were centrifuged at 20,680× *g* speed using Sorvall RC 6 plus Centrifuge (Thermo Scientific, Helsinki, Finland) for 45 min at 4 °C. The supernatants were filtered through a 0.22 µm polyethersulfone (PES) minisart syringe filter (Sartorius, Goettingen, Germany). The poly-prep chromatography columns (Bio-Rad, Pleasanton, CA, USA) were packed with the chromatography medium containing TALON Superflow matrix (GE Healthcare, Stockholm, Sweden) prepared in a 1:1 volume ratio of 1 mL slurry and 1 mL milli-Q water followed by the addition of 1 mL of 50 mM cobalt chloride (CoCl_2_). The cleared soluble fractions were loaded onto the packed column and then incubated for 1 h at 4 °C. The flow-through was discarded, and the column was pre-equilibrated with the binding buffer (50 mM Na_2_HPO_4_, 300 mM NaCl, pH 7.4) and washed with the wash buffer (50 mM Na_2_HPO_4_, 10 mM imidazole, 300 mM NaCl, pH 7.4), and finally, proteins were eluted using the elution buffer (50 mM Na_2_HPO_4_, 300 mM imidazole, 300 mM NaCl, pH 7.4).

### 4.3. FABP3 Purification Using Size-Exclusion Chromatography (SEC)

Size-exclusion chromatography (SEC) purification was performed with Superdex 75 increase 10/300 GL column (GE Healthcare Life Sciences, Stockholm, Sweden) using the ÄKTA Avant 25 chromatography system. The system was prepared for manual run with 1xphosphate-buffered saline (PBS), (10 mM Na_2_HPO_4_, 140 mM NaCl, 2.7 mM KCl, pH 7.4) running buffer (A1), milli-Q water (A2), and 20% ethanol column storage buffer (B3), followed by the sample injection into the system. Following the manufacturer’s instructions, the eluents were collected into the deep 96-well plates at a flow rate of 0.4 mL/min and 0.5 mL fraction volume per well.

### 4.4. Western Blot Analysis

Aliquots (500 µL) of bacterial cell suspension were collected and centrifuged after the 0.1 mM IPTG induction at different time points. Each cell pellet was lysed in 80 µL of 1 × SDS-PAGE buffer (150 mM Tris-HCl, 1.2% SDS, 30% glycerol, pH 6.8). Protein concentrations were determined using the Pierce BCA protein assay kit (Thermo Scientific). Cell extracts were diluted in equivalent amounts in 2 × Laemmli sample buffer (62.5 mM Tris-HCl, 2% SDS, 25% glycerol, pH 6.8), supplemented with 15% β-mercaptoethanol (Sigma). Samples were boiled at 95 °C for 5 min, and 10 µg of each sample was loaded on a 4–12% SDS-polyacrylamide gel (Invitrogen). The protein was transferred to a polyvinylidene fluoride (PVDF) membrane (Bio-Rad) in a 1 × transfer buffer (20% Bio-Rad 5× transfer buffer stock and 20% ethanol) by a Transblot Turbo device (Bio-Rad) following manufacturer’s instructions. The membrane was washed in blocking buffer (5% BSA in 0.1% TBS-Tween) and probed with primary antibodies diluted in blocking buffer (His_5_ anti-mouse, 1:1500; goat polyclonal anti-GFP, 1:2000; rat anti-FABP3, 1:2000; and anti-mouse β-tubulin, 1:5000) and incubated at 4 °C overnight. After 3× washes in 0.1% TBS-Tween, the membrane was probed with horseradish peroxidase (HRP)-conjugated secondary antibodies at 1:2000 dilution in blocking buffer and incubated at room temperature for 2 h. See more details on the antibodies under the antibody section. After final washes, protein expression was visualized using the Pierce ECL Western Blot kit (Thermo Scientific).

### 4.5. Coomassie Blue Staining

After purification, the FABP3 protein concentration was quantified using the Bradford protein assay kit (Bio-Rad) and analyzed by Coomassie Blue G-250 stain (Thermo Scientific). The protein expression was analyzed both in the soluble (supernatant and fractionated eluent) and the insoluble fractions of the total cell lysates using SDS-PAGE. An equal amount of each of the soluble samples (100 µL) and 2 × Laemmli sample buffer (LSB) (100 µL) supplemented with 15% β-mercaptoethanol (Sigma) were mixed and boiled at 95 °C for 5 min. For the insoluble samples, an aliquot of the total lysate was centrifuged to pellet the insoluble proteins. Both the total lysate and pellet were prepared by adding 2 × LSB and 1 × LSB, respectively, and boiled for 5 min. Soluble samples (10 µL), total cell extracts (10 µL), and the insoluble pellet sample (5 µL) were loaded onto a 4–20% polyacrylamide gel (Invitrogen). Gels were stained with Coomassie Blue. FABP3 was detected at the 17.6 kDa expected size.

### 4.6. Solid-Phase Peptide Synthesis

Peptides were synthesized manually by using Fmoc/tBu solid-phase chemistry. Fmoc-Rink-MBHA resin (0.56 mmol/g) (Iris Biotech GmbH, Marktredwitz, Germany) was placed into a polypropylene syringe fitted with a polyethylene filter disk. The resin was swollen with CH_2_Cl_2_ (1 × 20 min) and DMF (1 × 20 min). Fmoc was removed using a mixture of piperidine/DMF (3:7, 2 + 10 min). Coupling of the corresponding Fmoc-amino acids (4 equiv, Iris Biotech GmbH) was performed using *N,N’*-diisopropylcarbodiimide (DIC, 4 equiv, Iris Biotech GmbH) and ethyl cyano-(hydroxylimino)-acetate (Oxyma, 4 equiv, Iris Biotech GmbH) in DMF at room temperature for 3 h. Completion of the reaction was monitored by the Kaiser test [39]. After each coupling and deprotection step, the resin was washed with DMF (6 × 1 min) and CH_2_Cl_2_ (1 × 1 min). After completion of the synthesis, the peptide was cleaved with TFA/DTT/H_2_O/TIS (88:5:5:2) for 2 h at room temperature. The TFA was removed, and the peptide was precipitated with cold Et_2_O. The crude precipitate was dissolved in H_2_O/ACN and purified by reversed-phase HPLC (Merck-Hitachi, Darmstadt, Germany).

TAMRA was conjugated on resin at the *N*-terminus of the peptides. After Fmoc cleavage, 5(6)-carboxymethylrhodamine (TAMRA, 2.0 equiv, Sigma, Darmstadt, Germany), 1-hydroxy-7-azabenzotriazole (HOAt, 2.1 eq, Genscript, Piscataway Township, New Jersey, USA), and DIC (1.95 eq) were dissolved in DMF (10 mL/mmol TAMRA), preincubated for 10 min at room temperature, and added to the corresponding swollen peptide-resin for 16 h. The resin was washed with DMF and DCM until the filtrate was colorless. The peptide was cleaved from the resin with TFA/DTT/H_2_O/TIS (88:5:5:2) during 1 h at room temperature. TFA was removed, and the crude peptide was purified by preparative RP-HPLC (Merck-Hitachi).

### 4.7. Liquid Chromatography—Mass Spectrometry (LC-MS)

LC-MS was conducted using a 1200 series system (Agilent, Waldbronn, Germany) consisting of an autosampler, degasser, binary pump, column oven, and diode array detector coupled to a 6220 accurate-mass ToF-MS (Agilent). A Hypersil Gold C_18_ (150 mm × 2.1 mm, 3 μm particle size) was used as a column. Eluent A: H_2_O/CH_3_CN/HCO_2_H = 95/5/0.1, and eluent B: H_2_O/CH_3_CN/HCO_2_H = 5/95/0.1. The flow rate was 300 µL/min. The gradient was linear from 100% A to 98% B in 10 min; 1 min 98%; linear from 98% B to 100% A in 0.5 min; and 3.5 min at 100% A.

### 4.8. High Resolution Mass Spectrometry

High resolution mass spectra were recorded using a 6220 accurate-mass TOF LC/MS (Agilent). A Hypersil Gold C_18_ column (50 mm × 2.1 mm, 1.9 μm particle size) was used for the LC separation. The same solvents as for the HPLC-MS were used, and a linear gradient from 0 to 98% B over 4 min. was employed. The mass spectrometer was externally calibrated using Agilent tuning mix prior to measurement.

### 4.9. Preparative Reversed Phase—High Performance Liquid Chromatography (RP-HPLC)

Preparative RP-HPLC was performed using a Merck-Hitachi unit (controller: d-7000, pump: L7150, detector: L7420, detection wavelength λ = 220 nm).

Eluent A: H_2_O/CH_3_CN/TFA = 95/5/0.1, and eluent B: H_2_O/CH_3_CN/TFA = 5/95/0.1. The column was Macherey-Nagel Nucleosil C_18_ (250 mm × 21 mm, 10 μm particle size). The flow rate was 10 mL/min. The gradient was 2 min at 100% A; linear from 100% A to 100% B in 33 min; 5 min at 100% B; and linear from 100% B to 100% A in 5 min.

### 4.10. Surface Plasmon Resonance

Binding experiments were carried out in the PBS-Tween running buffer (10 mM Na_2_HPO_4_, 140 mM NaCl, 2.7 mM KCl, 0.05% Tween-20, pH 7.4). A-CooP (ACGLSGLGVA) or a scrambled control peptide (CVAALNADG) was dissolved in immobilization buffer (10 mM sodium acetate, pH 4.5). Following the Biacore T100 (GE Healthcare Bio-Sciences) manufacturer’s protocols, 400 mM EDC (1-ethyl-3-(3-dimethylaminopropyl) carbodiimide, and 100 mM NHS (*N*-hydroxysuccinimide) coupling reagent mixture in a 1:1 ratio was first loaded to activate the sensor chip surface at a flow rate of 10 µL/min for 7 min. The A-CooP peptide was immobilized on the CM5 sensor chip surface flow cell (FC-4), and the control peptide was immobilized on the reference surface flow cell (FC-3) via amine coupling reaction. Upon A-CooP and control peptide immobilization by a 10 min injection of the 2 mg/mL of each at a flow rate of 10 µL/min, 1 M ethanolamine-HCl (pH 8.0, GE Healthcare Bio-Sciences) was loaded to block the excess amine-reactive NHS-esters. Two-fold serial dilutions of the purified recombinant FABP3 (analyte) were prepared in the running buffer starting with the highest concentration (32 µM) to the lowest concentration (0.5 µM) and were placed in the Biacore T100 instrument. Binding assays were performed by continuous injections of the analytes into the flow cells at 25 °C temperature. Injection of each FABP3 diluent was set to a flow rate of 30 µL/min at an association time of 180 s and dissociation time of 300 s. The regeneration cycle was set to a flow rate of 10 µL/min for 30 s contact time. Steady state affinities were calculated from the reference-subtracted sensorgrams. The binding affinity was evaluated using the steady state affinity fit model.

### 4.11. Microscale Thermophoresis (MST)

FABP3 was labeled with a second-generation red amine-reactive NT-650-NHS fluorescent dye using the Nanotemper labeling kit (MO-L011; Nanotemper, https://nanotempertech.com/). The protein/dye concentration ratio and volume ratio were 1:3 and 1:1, respectively, in accordance with the manufacturer’s instructions. The actual concentration and degree of labeling (DOL) were determined using the extinction coefficient Ɛ_280_ = 18450 M^−1^ cm^−1^ for FABP3 and Ɛ_650_ =195000 M^−1^ cm^−1^ for the NT650 dye. Using the correction factor of 0.04 at 280 nm, the DOL for all labeling reactions was calculated according to manufacturer’s instructions. Binding experiments were carried out in a PBS-tween 20 buffer (10 mM Na_2_HPO_4_, 140 mM NaCl, 2.7 mM KCl, 0.05% tween 20, pH 7.4). The binding assay was performed with a fixed concentration (20 nM) of the fluorescently labeled FABP3 (target) and two-fold serially diluted decreasing concentrations (500 µM to 0.0153 µM) of the unlabeled A-CooP variants (ligand). The reaction mixture of 16 serial diluents of A-CooP or its analogues and FABP3 were filled into the capillary tubes (Nanotemper, MO-K022) and loaded in the Monolith NT 115 instrument accordingly. All experiments were carried out at 25 °C, 40% MST power (medium), and 60% LED power. Binding assay and measurements were performed in triplicates for each experimental setup.

### 4.12. Generation of the BT12-GFP Glioblastoma Cells

The generation of all glioblastoma patient-derived cells including BT12 cells have been previously reported [40]. GFP-expressing BT12 (BT12-GFP) cells were generated after co-transfection of GFP-encoding plasmid and lentiviral packaging plasmids CMVg and CMVΔ8.9 (Addgene, Teddington, London, UK) into the 293FT cells using Fugene6-transfection reagent (Promega). Subsequently, the virus supernatants were collected and filtered after 72 h. BT12 cells were transduced with the virus supernatants, and GFP expressing cells were stably selected by using 1 µg/mL doxycycline.

### 4.13. Cell Culture

U87MG (GFP or FABP3-GFP) cells were maintained with low glucose (1.0 g/L glucose) DMEM medium (Lonza, Walkersville, MD, USA) supplemented with 2 mM l-glutamine (Lonza), 1% penicillin/streptomycin (100 Units/mL penicillin and 100 µg/mL streptomycin, Lonza), and 10% fetal bovine serum (FBS, Biowest). BT12-GFP cells were maintained with the Dulbecco’s Modified Eagle’s Medium (DMEM)/F-12 (1:1) growth medium (Gibco) supplemented with 2 mM l-glutamine (Lonza), 1% penicillin/streptomycin (100 Units/mL penicillin and 100 µg/mL streptomycin, Lonza), 15 mM HEPES buffer (Lonza), 2% B27 (Gibco), 0.01 µg/mL recombinant human fibroblast growth factor (FBF-b, Peprotech), 0.02 µg/mL recombinant human epidermal growth factor (EGF, Peprotech), and 1 µg/mL doxycycline. 293FT cells were maintained with high glucose (4.5 g/L glucose) DMEM medium (Lonza) supplemented with 2 mM l-glutamine, 1% penicillin/streptomycin (100 Units/mL penicillin and 100 µg/mL streptomycin), and 10% fetal bovine serum (FBS). All cells were cultured in the humidified incubator at 37 °C under a 5% CO_2_ atmosphere.

### 4.14. In Vitro Studies

Human U87MG glioblastoma cells expressing GFP or FABP3-GFP were seeded at the density of 50,000 cells per well in a 24-well plate and grown on 13-mm diameter round coverslips (Thermo Scientific) overnight. The growth medium was aspirated after 24 h and replaced with 500 µL fresh serum- and phenol-free medium FluoroBrite DMEM (Gibco) supplemented with 1% l-glutamine (Lonza) containing TAMRA-conjugated A-CooP variants or control peptide at a concentration of 1 µg/mL. After one hour, the medium was aspirated and cells were washed twice with PBS. Cells were fixed with 4% paraformaldehyde (PFA) 0.1 M pH 7.4 (Histolab) for 10 min, washed with PBS, and stained with DAPI (Vectashield, Vector Laboratories) to visualize the nuclei. The coverslips were mounted on glass slides for imaging. We used low passaged (p1), fluorescence-activated cell sorted (FACS) and validated batches of human U87MG glioblastoma cells expressing GFP or FABP3-GFP for this experiment. The protein expression was verified by Western Blot analysis. The generation of the stable cell lines have been reported elsewhere [2].

### 4.15. Antibodies

Primary antibodies used for the Western Blot analyses include goat polyclonal anti-GFP (ab6673; Abcam, Cambridge, UK), monoclonal rat anti-FABP3 (MAB1678; R&D Systems, Abingdon, UK), anti-mouse β tubulin (556321; BD Pharmingen Biosciences, Vantaa, Finland), and His_5_ antibody (34660; Qiagen, Amtsgericht Dusseldorf, Germany). All the horseradish peroxidase (HRP)-conjugated secondary antibodies including polyclonal rabbit anti-mouse, rabbit anti-rat, and rabbit anti-goat were purchased from Dako (Santa Clara, CA, USA).

### 4.16. Peptides

Table 1 provides a list of all peptide analogues and the selected TAMRA conjugated versions. Both the non-labeled and TAMRA-labeled control peptides with scrambled sequence were purchased from United BioSystems Inc (www.unitedpeptide.com).

### 4.17. In Vivo Studies

All animal experiments were conducted according to the ethical guidelines of the provincial government of Finland under the project license No. ESAVI/403/2019. Five-week-old immunocompetent NMRI nude mice (Janvier, France) were intracranially engrafted with patient-derived BT12-GFP cells (50,000 cells/5 µL) using the stereotactic instrument. After 25 days of tumor growth, the glioblastoma-bearing mice were intravenously injected with 100 µL (in 0.9% saline solution) of 100 µM fluorescent TAMRA-A-CooP variants or the TAMRA-conjugated control peptide. The peptide was allowed to circulate for 1 h while the animal was under deep anesthesia (ketamine and xylazine cocktail at 100 mg/kg and 5 mg/kg, respectively). The peptide still in circulation was removed by heart perfusion using 10 mL saline and 10 mL 4% PFA. The brain and control organs were collected and post-fixed overnight at 4 °C, followed by thorough PBS washes. The organs were soaked in 15% and 30% sucrose solutions for 1–2 days each until the organs were fully immersed in solution. The sucrose-preserved organs were snap frozen in 2-methyl butane filled with dry ice and stored at −80 °C until sectioning. The brain and other organs were immersed in the Optimal Cutting Temperature (OCT) cryomount embedding medium (Tissue-Tek, Histolab) and sectioned into 9 µm tissue slices. Frozen sections were mounted onto superfrost plus microscope glass slides (25 × 75 × 1.0 mm, Thermo Scientific) and stained with DAPI to visualize the nuclei.

### 4.18. Imaging and Image Analyses

All images were acquired using the Zeiss Axio Imager Z2 epifluorescence widefield microscope. For the whole brain sections, images were generated using 3DHISTECH Panoramic 250 FLASH II digital slide scanner at the Genomic Biology Unit (https://www.helsinki.fi/en/researchgroups/genome-biology-unit/scanning). For the quantitative analysis of the acquired images, mean fluorescent intensity measurements were performed using the cell profiler image analysis tool (https://cellprofiler.org). A cell profiler pipeline for the measurement of the in vitro/in vivo TAMRA mean intensity was created.

### 4.19. Statistical Analysis

All quantitation and analyses were performed using Prism 8.3.0 (GraphPad Software, San Diego CA, USA). MST data were analyzed using the nonlinear regression curve fit with the logarithm (dose, x) vs. response (variable slope, y) equation. Dose-response curves follow the four-parameter dose-response (4PL) sigmoidal slope. MST and SPR data were expressed as mean ± SD for three independent experiments. The in vivo TAMRA mean intensity measurement was expressed as mean ± SEM between peptide groups (at least 3 animals per group), and each data point corresponded to one histologic section. In vivo data were analyzed by one-way ANOVA using post hoc Sidak test. In vitro TAMRA intensity measurement was expressed as mean ± SEM for 63 microscope images per peptide from three independent experiments. In vitro data were analyzed by two-way ANOVA using post hoc Tukey’s test.

## 5. Conclusions

In summary, we have performed alanine scanning to create glioblastoma homing peptide A-CooP variants and have analyzed their binding to the target protein FABP3 by using MST. We validated the results by both in vitro and in vivo glioblastoma models. Identification of a cysteine and two glycine residues in the CooP sequence could provide vital information for drug-conjugate design and development.

## Figures and Tables

**Figure 1 cancers-12-01836-f001:**
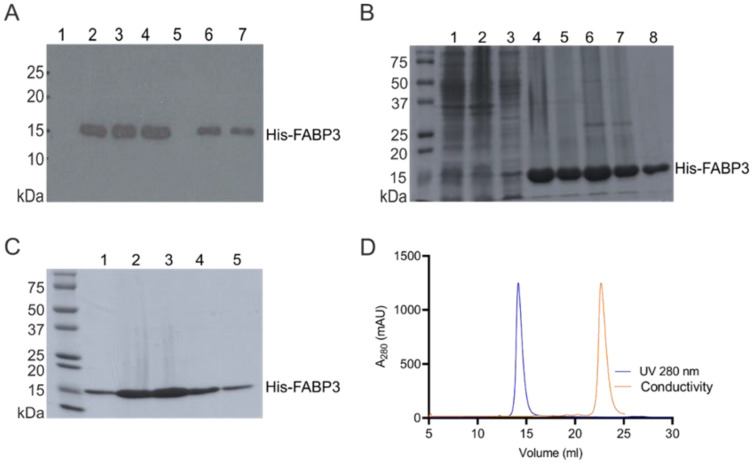
Recombinant human FABP3 expression, production, and purification: (**A**) Western Blot analysis of FABP3 expression after the isopropylthiogalactoside (IPTG) induction. Lanes 1 and 5 correspond to non-induced bacterial cultures, while lanes 2, 3, 4, 6, and 7 correspond to the IPTG-treated bacterial cultures. (**B**) Purification of N-terminal His-tagged FABP3 by immobilized metal affinity chromatography (IMAC): Lanes 1, 2, and 3 show the total cell extract, cell pellet, and the soluble extract, respectively. Lanes 4, 5, 6, 7, and 8 show fractionated eluents. (**C**) Consecutive purification of N-terminal His-tagged FABP3 by size exclusion chromatography (SEC). (**D**) Chromatogram shows two peaks: the blue peak (UV280 nm) corresponds to the purified His-tagged FABP3, and the yellow peak (conductivity) corresponds to impurities in the both Bradford protein assay and Coomassie Blue stain. FABP3 concentration was quantified after purification with a BioSpec nano spectrophotometer using the extinction coefficient (Ɛ) = 18,450 M^−1^ cm^−1^ and molecular mass = 17,592.03 g/mol at 280 nm. Quantitation showed a total yield of 10 mg/mL purified FABP3 after three consecutive runs of size exclusion chromatography.

**Figure 2 cancers-12-01836-f002:**
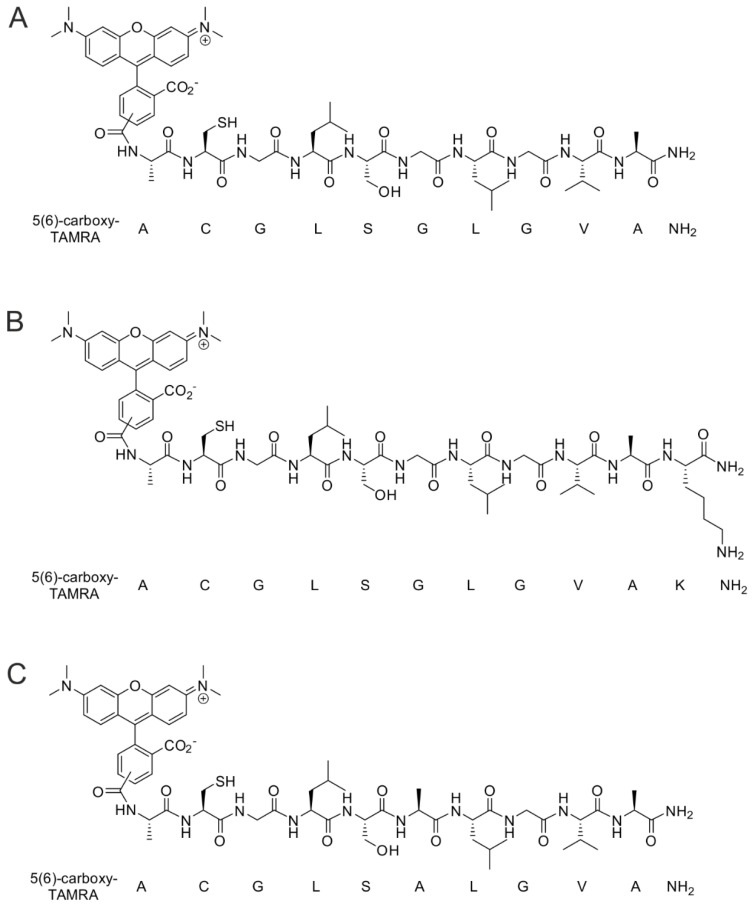
Synthesis of the selected A-CooP variants conjugated to fluorescent TAMRA dye: (**A**) A-CooP, (**B**) A-CooP-K, and (**C**) A-[Ala^5^]CooP. The sequence starts from the *N*-terminus to *C*-terminus. The *N*-terminus is modified by conjugation with TAMRA dye, and the *C*-terminus is modified by replacement of carboxylic acid with an amide group.

**Figure 3 cancers-12-01836-f003:**
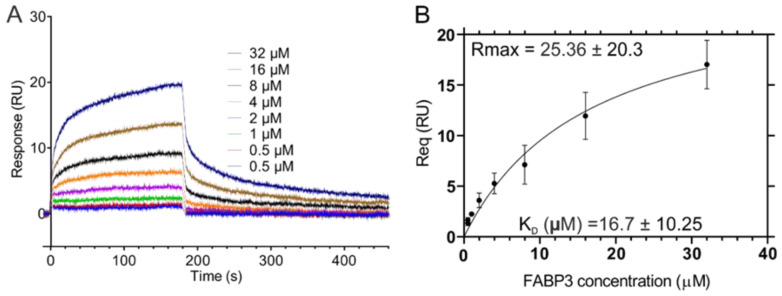
Measurement of the binding affinity between FABP3 and A-CooP peptides by using the surface plasmon resonance (SPR): (**A**) A representative SPR sensogram (n = 3) and (**B**) the mean steady state affinity between FABP3 and A-CooP peptide based on three independent experiments.

**Figure 4 cancers-12-01836-f004:**
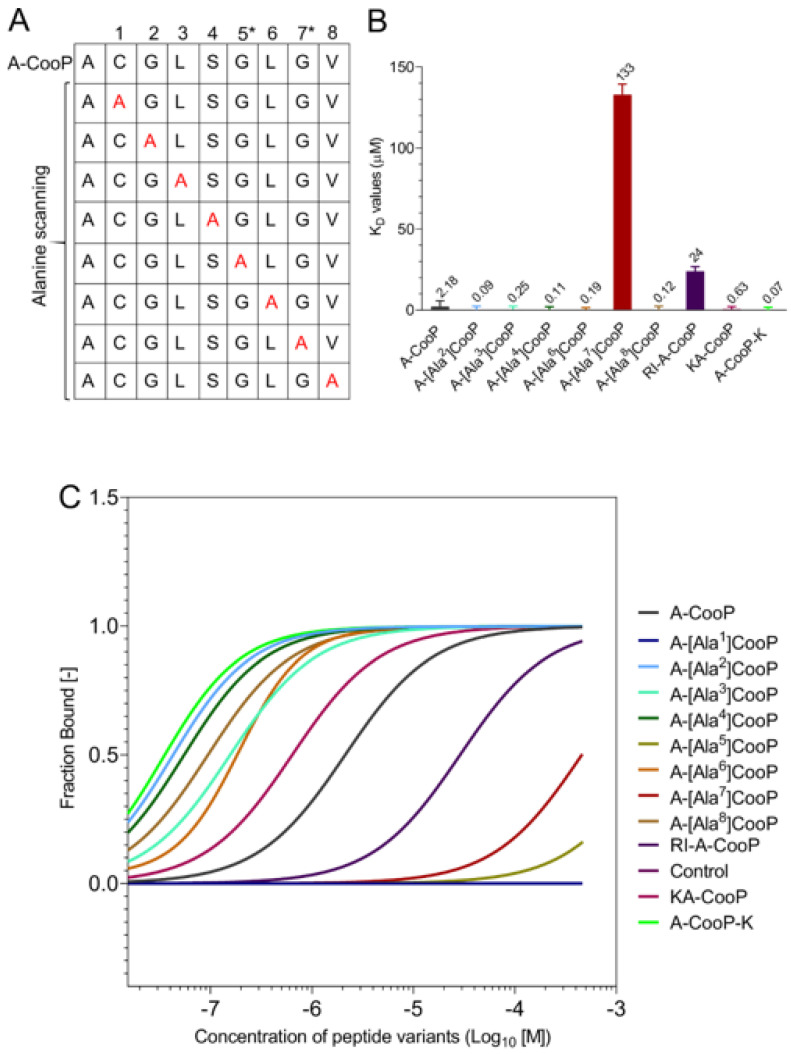
(**A**) The scheme shows the A-CooP peptide sequence and the variants with alanine residues (red) in each position of the peptide. (**B**) The bar graph shows the calculated K_D_ values for binding of the A-CooP variants to the FABP3 measured by microscale thermophoresis (MST). (**C**) Dose response curves of all the calculated fit values of the CooP variants upon binding to FABP3.

**Figure 5 cancers-12-01836-f005:**
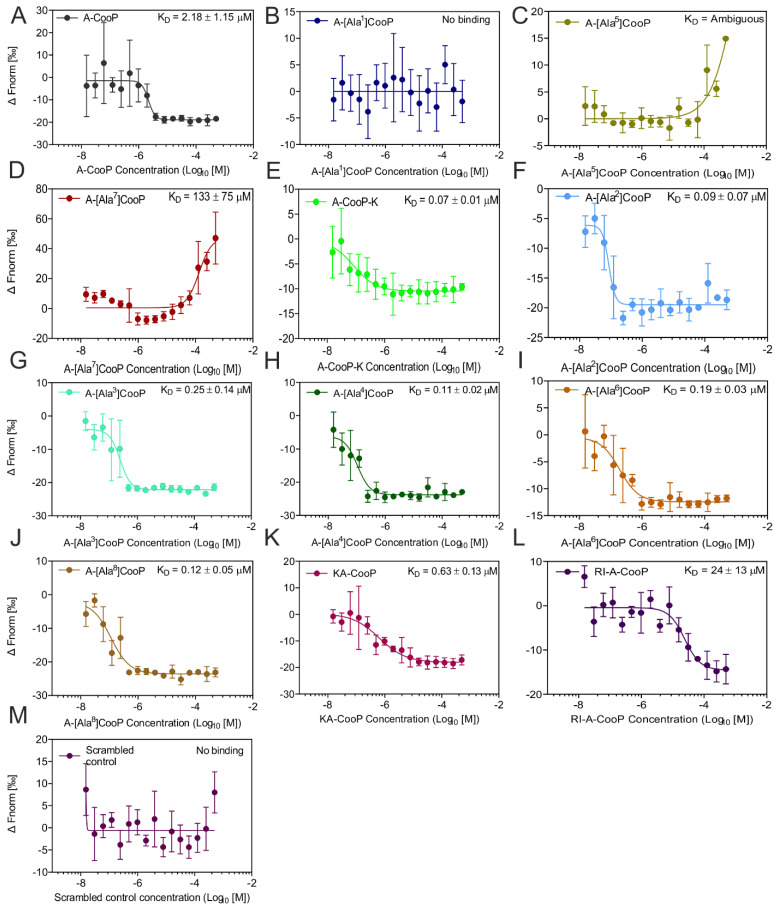
MST affinity curves for the binding of A-CooP variants to FABP3: (**A**–**M**) The x-axis shows the peptide concentration, and the y-axis shows the change in normalized fluorescence. Error bars represent the standard error of data points calculated from three independent experiments.

**Figure 6 cancers-12-01836-f006:**
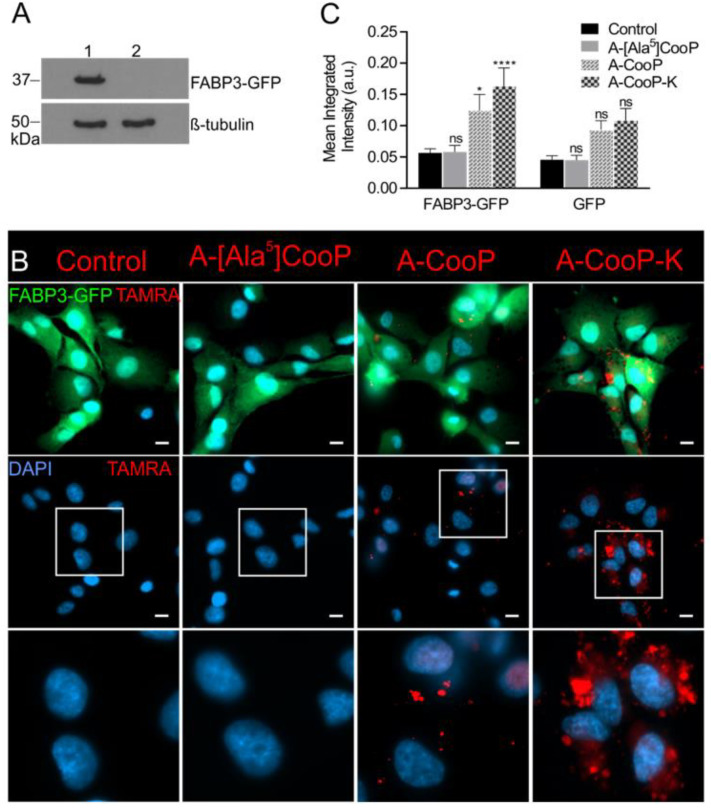
In vitro cellular uptake of the selected A-CooP variants: (**A**) Western Blot analysis shows FABP3 protein expression in the U87MG-FABP3-GFP cells (lane 1), while no expression is seen in the GFP-expressing U87MG cells (lane 2). (**B**) Microscopic images show internalization of the A-CooP-K and A-CooP peptides (red) in the U87MG glioblastoma cells expressing FABP3-GFP. A-[Ala^5^]CooP and the control peptide showed no binding or internalization to these cells. FABP3-GFP expressing tumor cells are seen in the upper row in green, and 4′,6-diamidino-2-phenylindole (DAPI) was used to visualize the nuclei (blue, middle and lower row). Higher magnification of the boxed areas in the images in the middle row are shown in the panels below. (**C**) The graph shows the quantitation of TAMRA mean integrated intensities of internalized A-CooP variants. Analysis was performed on 63 microscopic images per peptide from three independent experiments. Error bars represent the standard error of the mean (SEM). **** *p* < 0.0001 (A-CooP-K), * *p* = 0.0312 (A-CooP), and ns (A-[Ala^5^]CooP) when compared to control peptide in U87MG glioblastoma cells expressing FABP3-GFP. Statistical analysis was evaluated by 2-way ANOVA with Tukey’s post hoc test. Scale bar = 10 µm.

**Figure 7 cancers-12-01836-f007:**
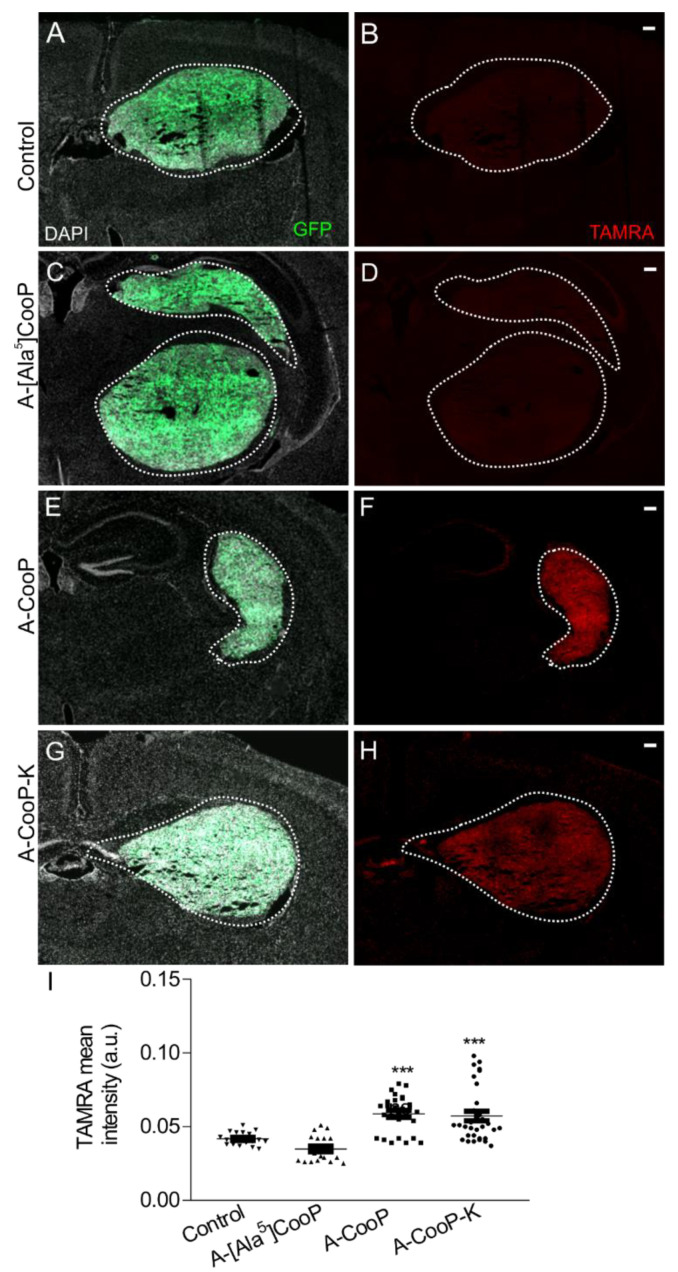
Homing of the A-CooP variants to the intracranial glioblastoma xenografts: Micrographs show the intracranial xenographs in green (outlined with white dashed lines) and the peptide-associated fluorescence in red: (**A**,**B**) Control peptide, (**C**,**D**) A-[Ala^5^]CooP, (**E**,**F**) A-CooP, and (**G**,**H**) A-CooP-K. (**I**) Quantitation of the mean intensities of the peptide-derived fluorescence in the intracranial tumors after 60 min of systemic administration: Measurements were performed on at least 20 histologic sections from at least three animals per peptide variant. The lines show the standard error of the mean (SEM). *** *p* = 0.0003 (A-CooP-K), *** *p* = 0.0001 (A-CooP) when compared to the control peptide using one-way ANOVA with Sidak’s post hoc test. These panels are a higher magnification of representative whole coronal section micrographs of murine brain containing the tumors (outlined with white dashed line) that are shown in Appendix A.

**Table 1 cancers-12-01836-t001:** Synthesis of the different alanine variants of the A-CooP peptide: A-[Ala^1^]CooP–A-[Ala^8^]CooP, KA-CooP, A-CooP-K, retroinverso A-CooP peptides, and the TAMRA-labeled selected A-CooP variants.

Abbreviation	Sequence (N – C)	T_R_ (min)	Purity (%)	m/z Measured/Theoretical
A-CooP	ACGLSGLGVA	5.32	97	846.4497/846.4502
A-[Ala^1^]CooP	AAGLSGLGVA	5.13	>99	814.4781/814.4781
A-[Ala^2^]CooP	ACALSGLGVA	5.27	94	860.4686/860.4658
A-[Ala^3^]CooP	ACGASGLGVA	4.40	>99	804.4009/804.4032
A-[Ala^4^]CooP	ACGLAGLGVA	5.41	96	830.4531/830.4553
A-[Ala^5^]CooP	ACGLSALGVA	5.43	98	860.4696/860.4658
A-[Ala^6^]CooP	ACGLSGAGVA	4.26	99	804.4039/804.4032
A-[Ala^7^]CooP	ACGLSGLAVA	5.42	>99	860.4684/860.4658
A-[Ala^8^]CooP	ACGLSGLGAA	5.02	>99	818.4204/818.4189
KA-CooP	KACGLSGLGVA	4.60	>99	974.5442/974.5451
A-CooP-K	ACGLSGLGVAK	4.47	>99	974.5435/974.5451
RI-A-CooP	avglgslgca	5.09	>99	846.4534/846.4502
Negative control	CVAALNADG	Commercial source		
TAMRA-A-CooP	TAMRA-ACGLSGLGVA	5.95 and 6.14 (isomers)	88	1258.5958/1258.5925
TAMRA-A-CooP-K	TAMRA-ACGLSGLGVAK	5.18 and 5.37 (isomers)	96	1386.6860/1386.6875
TAMRA- A-[Ala^5^]CooP	TAMRA-ACGLSALGVA	5.87 and 6.07 (isomers)	94	1272.6089/1272.6082
TAMRA-Negative control	TAMRA-CVAALNADG	Commercial source		

Red marks indicates the Alanine (A) or Lysine (K) modification of the peptides.

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
