# Peer review of "Tumor-Targeting Peptides: The Functional Screen of Glioblastoma Homing Peptides to the Target Protein FABP3 (MDGI)"

_cancers, 2020, doi:10.3390/cancers12071836_

Round 1

Reviewer 1 Report

The authors report data of a SAR study providing very first interesting insights into the determinants which govern the interaction of the glioblastoma homing peptide CooP with its target protein FABP3. In particular, a variant of the initially identified 9-mer CooP, A-CooP-K, which was elongated C-terminally by an additional Lys, turned out to be a 30-fold more potent binder than A-CooP.  Moreover, the glioblastoma targeting is improved in comparison to the lead peptide A-CooP.

Here arises a question, that has not been answered by the authors adequately. What was the rational to implement the peptides KA-CooP and A-CooP-K into the SAR study? Obviously, A-CooP-K was a good choice.

Furthermore, depending on the method (SPR or MST), the binding affinities for A-CooP differ by a factor of eight. To obtain here a more general picture, at least the binding affinities for A-CooP-K and A-[Ala5]CooP should be also determined by SPR and implemented in the manuscript.

In Addition, errors of the KD-values are missing in Figure 5, completely. These Errors have to be added.

Some further minor points have also to be addressed:

- Page 10, line 130: the citation format has to be switched from author name to numbering.

- Page 11, Figure 6 B: the readability of the inscriptions on the black background (in particular the red colored labels) is difficult. This has to optimized adequately.

- Page 11, Figure 6, legend to figure 6A: the lane numbers (1) and (2) from the blot have to be also mentioned in the text.

In conclusion, the data presented by the authors are a promising contribution to the field of tumor drug targeting. The study has been well performed and the obtained experimental data are well documented. Thus, the manuscript is recommended for publication in cancer after minor revision as indicated above.

Reviewer 2 Report

In this manuscript, Ayo and colleagues report a functional screen of the glioblastoma homing peptide CooP (CGLSGLGVA), which discovery has been reported in a previous paper by the same group. The Coop peptide was discovered through an in vivo phage display screen and it binds the mammary derived growth inhibitor (FABP3) in cancer cells.

Here, the authors performed a functional screen aimed at defining in vitro the binding characteristics of the peptide, e.g. amino acids critical for binding, affinity constants, etc, and its ability to target in vivo the tumoral tissue in a xenograft mouse model of glioblastoma.

Results from alanine scanning experiment show that cysteine in position 1, glycine in 5 and glycine in 7 are critical for binding, with the first two completely abolishing the binding capacity of the peptide when substituted with alanine. In vivo experiment in the mouse xenograft model demonstrated the ability of the peptide to target the tumoral tissue in vivo.

This is a nice paper, and results are presented concisely. I think it could be of interest to the broad readership of Cancers

Minor points:

The authors present A-CooP-K as an improved FABP3 binding peptide based on the superior binding properties in the MST and in vitro analyses. The rationale of this is not completely clear to me. On which basis did they decide to add a lysine, and how is it supposed to work? It would be helpful if the authors may add some more comments

Figure 3C: although not statistically significant, it looks that A-CooP and A-CooP-K peptides show some unspecific binding to GFP-transfected cells. The authors should add some comments on this.

Line 28: the term “payloads” looks out of context

Reviewer 3 Report

The paper supports the ability of A-Coop to home to glioblastoma cells in vitro and in vivo. 

It would benefit from reproduction in more than one cell line for the in vitro and in vivo experiments respectively. Furthermore the paper would be of much greater interest if a mechanism for crossing the blood brain barrier in the in vivo experiments could be supported. 

Reviewer 4 Report

MANUSCRIPT REVIEW

In the study entitled “Tumor-targeting peptides: The functional screen of glioblastoma homing peptides to the target protein FABP3 (MDGI)” Ayo et al. present an alanine scan of A-CooP to investigate the contribution of each amino acid residue to the binding to mammary-derived growth inhibitor (MDGI) (also known as FABP3 or H-FABP) by microscale thermophoresis (MST) and surface plasmon resonance (SPR). Also, the authors have tested the binding affinity of the A-CooP-K, KA-CooP, and the retro-inverso A-CooP analogues to the recombinant FABP3. According to the MST analysis, A-CooP showed micromolar affinity to FABP3. Furthermore, the results were corroborated in vitro and in vivo using glioblastoma models. In the opinion of the authors, the data provide insight into the FABP3-A-CooP interaction that may be relevant for future applications of drug conjugate design and development.

The work is interesting and several aspects are innovative. The research project and the experimental design are appropriate, the conceptional structure is well organized.

However, I recommend major revision because the Introduction needs to be improved, “Results” section show several information that can be moved in “Material and Methods” section and figures show some lacks. Also, not all relevant works have been listed in the “References” section.

Point by point 

INTRODUCTION

The conceptual structure of introduction is well organized but I suggest more accurate bibliography research because not all relevant works have been listed. In this way, you could develop this section for a better background.

RESULTS

In general, the “Results” section and the legends of the figures are full of informations that you should move in the “Materials and Methods” section (e.g. temperature of treatment, concentration of reagents).

Fig. 1

You should add the densitometric analysis of the Western Blot.

Line130. You should move the reference and change the format.

Figure 6

In Fig. 6A, you should add the densitometric analysis.

In Fig. 6B, the third column shows lack of information. Also, you should move the B letter out of the image.

MATERIALS AND METHODS

Line 262. You should provide other informations about Western Blot (e.g. blocking solution, primary antibody concentration, peroxidase, etc…).

Line 370. In the “In vitro studies” section, indicate if, when and how cell line was last authenticated, maximum number of passages before the cells were analyzed.

Line 383. You should move the thanksgiving in the acknowledgment section.

Line 392-394. A different font?

REFERENCES

You should add more references because not all relevant published works have been cited.

Author Response

Response to Comments and Suggestions for Authors

Reviewer 4

In the study entitled “Tumor-targeting peptides: The functional screen of glioblastoma homing peptides to the target protein FABP3 (MDGI)” Ayo et al. present an alanine scan of A-CooP to investigate the contribution of each amino acid residue to the binding to mammary-derived growth inhibitor (MDGI) (also known as FABP3 or H-FABP) by microscale thermophoresis (MST) and surface plasmon resonance (SPR). Also, the authors have tested the binding affinity of the A-CooP-K, KA-CooP, and the retro-inverso A-CooP analogues to the recombinant FABP3. According to the MST analysis, A-CooP showed micromolar affinity to FABP3. Furthermore, the results were corroborated in vitro and in vivo using glioblastoma models. In the opinion of the authors, the data provide insight into the FABP3-A-CooP interaction that may be relevant for future applications of drug conjugate design and development.

The work is interesting and several aspects are innovative. The research project and the experimental design are appropriate, the conceptional structure is well organized.

However, I recommend major revision because the Introduction needs to be improved, “Results” section show several information that can be moved in “Material and Methods” section and figures show some lacks. Also, not all relevant works have been listed in the “References” section.

Point by point 

INTRODUCTION

Q1. The conceptual structure of introduction is well organized but I suggest more accurate bibliography research because not all relevant works have been listed. In this way, you could develop this section for a better background.

We appreciate the reviewer’s insightful and thoughtful comments towards an improved manuscript. We have added few lines to the introduction to capture more relevant works in the field and we hope it now reads well to the audience. Please see page 1, lines 38-39, 44-45 and page 2, lines 1-2 in the revised manuscript.

RESULTS

Q2. In general, the “Results” section and the legends of the figures are full of informations that you should move in the “Materials and Methods” section (e.g. temperature of treatment, concentration of reagents).

We have moved this information from the ‘Results’ section to ‘Materials and methods’ section in the revised manuscript.

Q3. Fig. 1

You should add the densitometric analysis of the Western Blot.

While we appreciate the reviewer’s comments on this matter, we do not see the point for the densitometric analysis of the Western Blot. The purpose of the blot is to show the protein of the right molecular weight to be produced by the bacterial cultures after the induction by IPTG. The protein amount was then measured by BioSpec nano spectrophotometer.

Q4. Line130. You should move the reference and change the format.

We thank the reviewer for pointing out the citation format error. We have moved the reference and changed the format accordingly (see page 9, line 1 in the revised manuscript).

Q5. Figure 6

In Fig. 6A, you should add the densitometric analysis.

Also, in this Western Blot analysis we do not see the need for densitometric analyses. The purpose of this Western Blot is merely to show that the FABP3-GFP is expressed in the U87MG cells at high level compared to the GFP expressing U87MG cells.

Q6. In Fig. 6B, the third column shows lack of information. Also, you should move the B letter out of the image.

We have now revised the images so that the information would be more visible. The third column has been re-oriented to a row and the letter B has been moved to the top left bar and is no longer interfering with the image. The third row shows the higher magnification of the boxed area in images presented in the middle row to highlight the A-CooP and A-CooP-K peptide internalization by the cells.  Control and Ala5 show no signal because the peptides were not internalized while A-CooP and A-CooP-K show moderate and excellent internalization, respectively.

MATERIALS AND METHODS

Q7. Line 262. You should provide other informations about Western Blot (e.g. blocking solution, primary antibody concentration, peroxidase, etc.…).

We have revised the ‘Western Blot analysis’ under ‘Material and Methods’ section to include the blocking buffer solution, primary and secondary antibody concentrations, IPTG concentration, washing buffers, antibody incubation time, and name of the blotting substrate used to visualize FABP3 expression and other applicable information (page 12, lines 4146 and page 13 lines 1-4 in the revised manuscript).

Q8. Line 370. In the “In vitro studies” section, indicate if, when and how cell line was last authenticated, maximum number of passages before the cells were analyzed.

We have used low passaged cell lines (p1) from the original and first batch of transduced cells sorted with BD LSR II fluorescence-activated cell sorter. The stably expressing cells have been selected by doxycycline and expression was confirmed by Western Blot analysis. The generation of the stable cell lines have been reported by earlier (Hyvönen et al Mol Cancer Ther, 2014). We have added this information in the “In vitro studies” section (page 15, lines 29 - 32 in the revised manuscript).

Q9. Line 383. You should move the thanksgiving in the acknowledgment section.

We have moved the thanksgiving previously in lines 383 and 225 to the acknowledgment section. (page 17, lines 11-13 in the revised manuscript).

Q10. Line 392-394. A different font?

We have changed the font type and size to Palatino Linotype and 10 pt accordingly.

REFERENCES

Q11. You should add more references because not all relevant published works have been cited.

We have updated the references and increased the number by 11 additional references. New references are numbers 8-10, 13-16 and 35-38.

Round 2

Reviewer 4 Report

The authors addressed the most of my concerns and so, about my opinion, the work is now suitable for publication in Cancers journal.